# *Pneumocystis jirovecii* Pneumonia Prophylaxis for Cancer Patients during Chemotherapy

**DOI:** 10.3390/pathogens10020237

**Published:** 2021-02-19

**Authors:** Kazuto Takeuchi, Yoshihiro Yakushijin

**Affiliations:** 1Cancer Center, Ehime University Hospital, Toon, Ehime 7910295, Japan; kazutake@m.ehime-u.ac.jp; 2Department of Clinical Oncology, Ehime University Graduate School of Medicine, Toon, Ehime 7910295, Japan

**Keywords:** *Pneumocystis jirovecii* pneumonia (PJP), cancer chemotherapy, prophylaxis, smoking

## Abstract

*Pneumocystis jirovecii* pneumonia (PJP) is one type of life-threatening pneumonia in immunocompromised patients. PJP development should be considered in not only immunocompromised individuals, but also patients undergoing intensive chemotherapies and immunotherapies, organ transplantation, or corticosteroid treatment. Past studies have described the clinical manifestation of PJP in patients during chemotherapy and reported that PJP affects cancer treatment outcomes. Therefore, PJP could be a potential problem for the management of cancer patients during chemotherapy, and PJP prophylaxis would be important during cancer treatment. This review discusses PJ colonization in outpatients during cancer chemotherapy, as well as in healthy individuals, and provides an update on PJP prophylaxis for cancer patients during chemotherapy.

## 1. Introduction

*Pneumocystis jirovecii* pneumonia (PJP) is an opportunistic infection caused by the yeast-like fungus PJ. PJP is a type of life-threatening pneumonia in immunocompromised patients including those with corticosteroid treatment, hematological and solid organ malignancies, organ transplantation, autoimmune disease, and human immunodeficiency virus. Two clinical factors need to be considered regarding the onset of this pneumonia; one is the airway environment, such as mucus damage from air pollution, chemical substances associated with cancer chemotherapy, and colonization of bacteria or fungi in the airway during cancer chemotherapy, in which PJ settles, and the other is host immunity against PJ infection after the administration of anti-tumor and immunosuppressive agents. PJ DNA is detectable in sputum or bronchoalveolar lavage specimens, not only from PJP and immunocompromised patients but also from healthy individuals, suggesting that PJ transmission may occur via an airborne route. As a result, PJ can colonize airways and pulmonary alveoli of some healthy individuals with latent infection. However, healthy individuals rarely develop PJP, even in the event of PJ colonialization. Environmental risk factors and host immunity are closely involved in PJP development.

In this review, we discuss the following:Mucosal damage and the risks of PJ colonization;Diagnosis of PJP;Host immunity-associated risks of PJP for patients during cancer chemotherapy;Chemoprophylaxis for PJP (first- and second-line) in immunocompromised patients.

## 2. Mucosal Damage and PJ Colonization

The normal mucosa of the throat and lower respiratory tract plays an important role in protecting the host from pathogenic microorganisms. Mucosal damage is often caused by several factors, such as respiratory infections, autoimmune diseases of the respiratory tract, and chemical substances after inhalation, aspiration, or medical treatments. One of the most hazardous chemical substances for human respiratory mucosa is tobacco smoke. In contrast, mucosal damage occurs from chemical substances associated with chemotherapy, including 5-fluorocytosine, 5-fluorouracil, cyclophosphamide, cisplatin, carboplatin, docetaxel, paclitaxel, and vinorelbine [1], which suggests that the lack of bronchial mucosa during chemotherapy may be a risk factor of bacterial or fungal colonization. In addition, myelosuppression during cancer chemotherapies may also promote the pathogen colonization in the tracheal mucosa.

PJ colonization in airways and air vesicles may develop after the destruction of anatomical barriers. Our group has examined PJ colonization via the airborne route using nested PCR with specific primers targeting the PJ gene (mitochondrial small subunit rRNA gene) among cancer patients during chemotherapy compared to healthy individuals. PJ DNA was detectable in 46% of sputum specimens from cancer patients during chemotherapy, which was not significantly different among cancer types and chemotherapy regimens, and the prophylactic use of trimethoprim/sulfamethoxazole (TMP/SMX) reduced the detection of PJ DNA (Table 1) [2]. Interestingly, PJ DNA was detected at a higher rate in healthy smokers (47%) compared with healthy non-smokers (20%), suggesting that smoking may be associated with PJ colonization in airways and air vesicles and may increase the mortality rate of PJP among cancer patients. Destruction of the mucosal barrier of the throat and lower respiratory tract with anticancer agents and air pollution may induce PJ colonization, and these processes may be involved in PJP development as a first step.

## 3. Diagnosis of PJP

The diagnosis of PJP requires microbiological tests and radiological findings. The following features of PJP have been reported.

### 3.1. Microbiological Tests

Microscopic examination of respiratory specimens (oral washes, expectorated or induced sputum, tracheal secretions, and broncho-alveolar lavage (BAL)) using various staining methods, such as Giemsa stains or direct and indirect immunofluorescent assays, has been used to visualize and identify the morphological structures of PJ. Staining methods have now largely been supplanted by highly sensitive molecular techniques, using semi- or fully quantitative polymerase chain reaction (PCR) targeting PJ-specific genes. Quantitative PCR (qPCR), with defined upper- and lower-quantitation thresholds of the PJ copy number, can be used to distinguish true infection from colonization.

In addition, serological examinations of fungal infection, such as wall polysaccharide (1-3)-beta-d-glucan (BDG) of fungi, may be helpful. However, cross-reactions with certain hemodialysis filters, beta-lactam antimicrobials, and immunoglobulins, which raise concerns about false-positives, should be considered.

### 3.2. Radiological Findings

Chest radiographs (chest X-ray) in patients with PJP often show small pneumatoceles, subpleural blebs, and fine reticular interstitial changes that are predominantly perihilar in distribution. Pleural effusions are normally not a feature. The most frequent CT findings are bilateral, ground-glass changes with apical predominance and peripheral sparing. The range of other radiological features seen in PJP includes a combination of ground glass and consolidative opacities, cystic changes, linear-reticular opacities, solitary or multiple nodules, and parenchymal cavities. In addition, nuclear imaging modalities such as 18F-fluorodeoxyglucose-positron emission tomography (FDG-PET) have been used as an adjunct to plain X-ray or CT. FDG-PET shows bilateral uptake of FDG in the upper zones of the lungs.

## 4. Host Immunity-Associated Risks for PJP

Several risk factors of PJP concerning host immunity have been reported. An increased risk of developing PJP has been reported for patients with T lymphocyte-mediated immunodeficiency with fewer than 200 CD4^+^ T cells/μL [3]. Intermittent administration of corticosteroid, which is often used in chemotherapy, accumulates and poses a risk of PJP development because of cell-mediated immunodeficiency due to the depletion of T lymphocytes in the circulatory system [4]. The daily dose and duration of corticosteroids, effects of immunosuppressive and anticancer agents used in combination with corticosteroids, and effects of underlying diseases are very diverse. However, as often stated, PJP prophylaxis can be considered in patients receiving the prednisolone (PSL) equivalent of 20 mg or more daily for four or more weeks [5].

## 5. Patients in Need of PJP Prophylaxis during Cancer Chemotherapy

Patients with hematological malignancies and solid tumors are the most common population in need of PJP prophylaxis. In addition, several reports have shown higher risks of developing PJP in hematopoietic stem cell or organ transplant patients [4,6,7,8]. Patients at the highest risk of PJP are considered to be those undergoing treatment for allogeneic hematopoietic stem cell transplant (HSCT) and acute lymphoblastic leukemia (ALL). PJP prophylaxis should be administered in allogeneic HSCT patients for at least six months, and while receiving immunosuppressive therapy. ALL patients should receive PJP prophylaxis throughout anti-leukemic therapy. These diseases and PJP prophylaxis are stipulated as Category 1 in the NCCN guidelines [5]. Patients treated with alemtuzumab should receive PJP prophylaxis for a minimum of two months after alemtuzumab treatment, and until thw CD4 count is greater than 200 cells/μL. Additionally, patients treated with phosphatidylinositol-3-kinase (PI3K) inhibitors (copanlisib and idelalisib [9]) and/or rituximab should receive PJP prophylaxis at least through active treatment. PJP prophylaxis should also be considered in patients receiving the prednisolone (PSL) equivalent of 20 mg or more daily for four or more weeks, at least through active treatment. PJP prophylaxis should be used when temozolomide is administered concomitantly with radiation therapy and should be continued until recovery from lymphocytopenia. Recipients of purine analog therapy and other T cell-depleting agents should receive PJP prophylaxis until CD4 count is greater than 200 cells/μL. PJP prophylaxis can be used in autologous HSCT patients for up to 3–6 months after transplant.

PJP prophylaxis should be considered for the following upcoming molecular-targeted drugs [9]: Bruton’s tyrosine kinase inhibitors (ibrutinib, acalabrutinib, and zanubrutinib), BCR-ABL tyrosine kinase inhibitors (imatinib, bosutinib, dasatinib, nilotinib, and ponatinib), PI3K inhibitors (copanlisib and idelalisib), mTOR inhibitors (everolimus, sirolimus, and temsirolimus), and Janus kinase inhibitor (ruxolitinib). In addition, PJP prophylaxis can be considered for the following monoclonal antibodies and fusion proteins [10,11]: blinatumomab, brentuximab, daratumumab, and alemtuzumab (for CD4 count <200 cells/μL). Moreover, patients treated with chimeric antigen receptor-engineered T cell therapy (axicabtagene ciloleucel for large B cell lymphoma, tisagenlecleucel for B-ALL) should receive PJP prophylaxis. 

## 6. Chemoprophylaxis for PJP

Chemoprophylaxis for PJP (Table 2).

### 6.1. Trimethoprim/Sulfamethoxazole (TMP/SMX)

TMP/SMX is the first-line drug for PJP prophylaxis, which is widely used in adults and children, and the only drug that has confirmed efficacy in prospective randomized controlled trials [12,13]. The optimal dose of TMP/SMX is not clear because of the limited number of studies in cancer patients during chemotherapy [12,13,14]. However, for adults, 80 mg/400 mg or 160 mg/800 mg TMP/SMX daily or 160 mg/800 mg TMP/SMX three times a week is recommended [5,6,7,15]. For children older than one month, 75 mg/m^2^/dose or 2.5 mg/kg/dose (max 160 mg/dose) TMP/SMX three times a week every 12 h is recommended [6]. All patients should be aware of adverse side effects such as hypersensitivity, renal damage, and myelosuppression, and TMP/SMX use may be restricted due to these events. 

### 6.2. Second-Line Drugs 

No powered trials have provided sufficient evidence for PJP prophylaxis in patients during chemotherapy. However, the following drugs should be considered when TMP/SMX is not available.

Dapsone is a synthetic sulfone (diaminodiphenyl sulfone; DDS). DDS works by inhibiting dihydrofolic acid synthesis in PJ [16]. For adults, 100 mg DDS daily is recommended. For children aged one month and older, 2 mg/kg/dose DDS daily or 4 mg/kg/dose DDS once a week is recommended [6]. 

Pentamidine is an inhaled and intravenous drug. It works by inhibiting glucose metabolism, protein synthesis, and amino acid transport in PJ [17]. For adults, 300 mg pentamidine inhaled through a nebulizer (administered through a jet-nebulizer producing a droplet size of 1–2 microns) every four weeks is recommended. For children aged two years and older, 4 mg/kg/dose (max 300 mg) intravenous pentamidine every four weeks is recommended. For children older than five years, 300 mg pentamidine inhaled through a nebulizer (administered through a jet-nebulizer producing a droplet size of 1–2 microns) every four weeks is recommended [6].

Atovaquone is a compound belonging to the class of naphthoquinone derivatives. It works by inhibiting ubiquinone binding to *cytochrome b* in PJ mitochondria. For adults, 1500 mg atovaquone daily with a high-fat meal is recommended. For children aged 1–3 months, 30 mg/kg/dose atovaquone daily is recommended. For children aged 4–24 months, 45 mg/kg/dose (max 1500 mg) atovaquone daily is recommended. For children older than 24 months, 30 mg/kg/dose (max 1500 mg) atovaquone daily is recommended [6].

## 7. Conclusions

PJP is a life-threatening pneumonia caused by disease- and treatment-related immunosuppression in cancer patients during chemotherapy, and appropriate prophylaxis is important for these patients. In addition to conventional circumstances such as ALL, HSCT, and corticosteroid administration, the number of conditions requiring PJP prophylaxis, such as patients treated with targeted therapies or monoclonal antibodies, is increasing. TMP/SMX is one strategy for PJP prophylaxis, and several other medications may be considered depending on the patient’s condition. Air pollution may be associated with PJ colonization in the airway and air vesicles and may increase the mortality rate of PJP. All patients undergoing cancer chemotherapies should cease smoking.

## Figures and Tables

**Table 1 pathogens-10-00237-t001:**
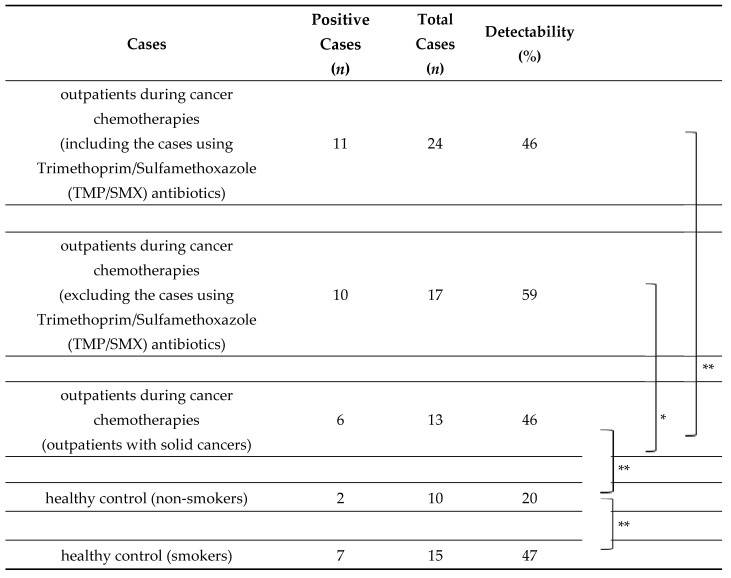
*Pneumocystis jiroveci* DNA detection from the sputum among outpatients during cancer chemotherapies and healthy controls (smokers or non-smokers) [2].

* *p* < 0.05, ** N.S.

**Table 2 pathogens-10-00237-t002:** Chemoprophylaxis for *Pneumocystis jiroveci* pneumonia [6].

	For Adults	For Children
TMP/SMX	80 mg/400 mg or 160 mg/800 mg daily or 160 mg/800 mg 3 times a week	For older than 1 month, 75 mg/m2/dose or 2.5 mg/kg/dose (max 160 mg/dose) 3 times a week every 12 h
Dapsone	100 mg daily	For aged 1 month and older, 2 mg/kg/dose daily or 4 mg/kg/dose once a week
Pentamidine	300 mg inhaled through a nebulizer every 4 weeks	For aged 2 years and older, 4 mg/kg/dose (max 300 mg) intravenously every 4 weeks For older than 5 years, 300 mg inhaled through a nebulizer every 4 weeks
Atovaquone	1500 mg daily	For aged 1–3 months, 30 mg/kg/dose daily For aged 4–24 months, 45 mg/kg/dose (max 1500 mg) daily For older than 24 months, 30 mg/kg/dose (max 1500 mg) daily

## Data Availability

Not applicable.

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
