# Peer review of "Pneumocystis jirovecii Pneumonia Prophylaxis for Cancer Patients during Chemotherapy"

_pathogens, 2021, doi:10.3390/pathogens10020237_

Round 1

Reviewer 1 Report

The review does not provide information on the pathogenic role  of PJ or the criteria for attributing pneumonia to PJ.

Pneumonia in an immunocompromised patient with positive PJ culture may not be due to PJ which may be a commensal?

Response rates to specific antiPJ treatment are needed

Author Response

February 2, 2021

To: Pathogens Editorial Office

 Thank you very much for your letter of Dec. 30, 2020 regarding our manuscript “Pneumocystis jirovecii pneumonia prophylaxis for cancer patients during chemotherapy.” (Manuscript ID: pathogens-1076176). We have responded to the helpful comments one by one and modified our manuscript as follows: Reviewer 1

#1. The review does not provide information on the pathogenic role of PJ or the criteria for attributing pneumonia to PJ.

Thank you very much for your helpful comments and suggestions.

According to this comment from Reviewer 1, we have added the diagnostic features of PJP to our manuscript, as follows:

Diagnosis of PJP

The diagnosis of PJP requires microbiological tests and radiological findings. The following features of PJP have been reported.

  1. Microbiological tests

Microscopic examination of respiratory specimens (oral washes, expectorated or induced sputum, tracheal secretions, and broncho-alveolar lavage [BAL]) using various staining methods, such as Giemsa stains or direct and indirect immunofluorescent assays, has been used to visualize and identify the morphological structures of PJ. Staining methods have now largely been supplanted by highly sensitive molecular techniques, using semi- or fully quantitative polymerase chain reaction (PCR) targeting PJ-specific genes. Quantitative PCR (qPCR), with defined upper- and lower-quantitation thresholds of PJ copy number, can be used to distinguish true infection from colonization.

In addition, serological examinations of fungal infection, such as wall polysaccharide (1-3)-beta-d-glucan (BDG) of fungi, may be helpful. However, cross-reactions with certain hemodialysis filters, beta-lactam antimicrobials, and immunoglobulins, which raise concerns about false-positives, should be considered.

  1. Radiological findings

Chest radiographs (chest X-ray) in patients with PJP often show small pneumatoceles, subpleural blebs, and fine reticular interstitial changes that are predominantly perihilar in distribution. Pleural effusions are normally not a feature. The most frequent CT findings are bilateral, ground-glass changes with apical predominance and peripheral sparing. The range of other radiological features seen in PJP includes a combination of ground glass and consolidative opacities, cystic changes, linear-reticular opacities, solitary or multiple nodules, and parenchymal cavities. In addition, nuclear imaging modalities such as 18F-fluorodeoxyglucose-positron emission tomography (FDG-PET) have been used as an adjunct to plain X-ray or CT. FDG-PET shows bilateral uptake of FDG in the upper zones of the lungs.

#2. Pneumonia in an immunocompromised patient with positive PJ culture may not be due to PJ which may be a commensal?

In some cases, early phases of PJ infection may be diagnosed as commensal or latent infection. However, the majority of PJ infections in immunocompromised patients are clinical pathogenic. That is why we have emphasized the significance of prophylactic administration as described in reference #2.

#3. Response rates to specific anti-PJ treatment are needed.

This comment is difficult to respond to because we do not have any randomized clinical trials to compare efficacies of prophylactic agents in immunocompromised patients. We would like to show the clinical dosage of each agent from the guidelines in this review. We are very sorry if we are unable to thoroughly address your comment.

Thank you very much for your helpful comments.

The detailed review of our manuscript is appreciated, and we have attempted to respond to all of the issues raised to the best of our ability. Thank you for your consideration of the revised version.

Sincerely,

Kazuto Takeuchi, M.D., Ph.D.

Yoshihiro Yakushijin, M.D., Ph.D.

Reviewer 2 Report

The review manuscript by Kazuto Takeuchi and Yoshihiro Yakushijin entitled “Pneumocystis jirovecii pneumonia prophylaxis for cancer patients during chemotherapy” summarized the recent progress of prevention of Pneumocystis jirovecii pneumonia in cancer patients who are under chemotherapy.

The introduction of mechanisms of Pneumocystis jirovecii pneumonia is oversimplified. The authors described mucus damage and colonization of the microbial in the airway, this part can be improved by adding more details.

In addition, a schematic graph summarizing the main idea of the manuscript may help readers to easily catch the major points.

I am not understanding the word use of “air vesicles” in Line 32, that phrase is specifically used in plant.

Author Response

February 2, 2021

To: Pathogens Editorial Office

 Thank you very much for your letter of Dec. 30, 2020 regarding our manuscript “Pneumocystis jirovecii pneumonia prophylaxis for cancer patients during chemotherapy.” (Manuscript ID: pathogens-1076176). We have responded to the helpful comments one by one and modified our manuscript as follows:

Reviewer 2

#1. The introduction of mechanisms of Pneumocystis jirovecii pneumonia is oversimplified. The authors described mucus damage and colonization of the microbial in the airway, this part can be improved by adding more details.

Thank you very much for your helpful comments and suggestions. We have added the following sentences to the Introduction:

Pneumocystis jirovecii pneumonia (PJP) is an opportunistic infection caused by the yeast-like fungus PJ. PJP is a type of life-threatening pneumonia in immunocompromised patients including those with corticosteroid treatment, hematological and solid organ malignancies, organ transplantation, autoimmune disease, and human immunodeficiency virus. Two clinical factors need to be considered regarding the onset of this pneumonia; one is the airway environment, such as mucus damage from air pollution, chemical substances associated with cancer chemotherapy, and colonization of bacteria or fungi in the airway during cancer chemotherapy, in which PJ settles, and the other is host immunity against PJ infection after the administration of anti-tumor and immunosuppressive agents. PJ DNA is detectable in sputum or bronchoalveolar lavage specimens not only from PJP and immunocompromised patients but also from healthy individuals, suggesting that PJ transmission may occur via an airborne route.

#2. A schematic graph summarizing the main idea of the manuscript may help readers to easily catch the major points.

Thank you very much your informative suggestion. We have added the following sentences after the Introduction.

In this review, we discuss the following:

  1. Mucosal damage and the risks of PJ colonization
  2. Diagnosis of PJP
  3. Host immunity-associated risks of PJP for patients during cancer chemotherapy
  4. Chemoprophylaxis for PJP (first- and second-line) in immunocompromised patients

#3. I am not understanding the word use of “air vesicles” in Line 32, that phrase is specifically used in plant.

We are very sorry for our incorrect explanation. We have changed “air vesicles” to “pulmonary alveoli” in the current article.

Thank you very much for your helpful comments.

The detailed review of our manuscript is appreciated, and we have attempted to respond to all of the issues raised to the best of our ability. Thank you for your consideration of the revised version.

Sincerely,

Kazuto Takeuchi, M.D., Ph.D.

Yoshihiro Yakushijin, M.D., Ph.D.

Round 2

Reviewer 1 Report

This appears to be an improved manuscript except for the misuse of "normal" which should be replaced by "usual"

Reviewer 2 Report

I do not have further concerns.